# A Low-Cost Underground Garage Navigation Switching Algorithm Based on Kalman Filtering

**DOI:** 10.3390/s19081861

**Published:** 2019-04-18

**Authors:** Ningbo Li, Yanbin Gao, Ye Wang, Zhejun Liu, Lianwu Guan, Xin Liu

**Affiliations:** 1Collage of Automation, Harbin Engineering University, Harbin 150001, China; liningbo2016@hrbeu.edu.cn (N.L.); yanbinbo9988@126.com (Y.G.); wangyeheu@hotmail.com (Y.W.); lzj1436791649@outlook.com (Z.L.); 2School of Aeronautics Engineering, Harbin Institute of Technology, Harbin 150006, China; 15B904027@hit.edu.cn

**Keywords:** parking lot, positioning, indoor and outdoor switching navigation method

## Abstract

Modern parking lots have gradually developed into underground garages to improve the efficient use of space. However, the complex design of parking lots also increases the demands on vehicle navigation. The traditional method of navigation switching only uses satellite signals. After the Position Dilution Of Precision (PDOP) of satellite signals is over the limit, vehicle navigation will enter indoor mode. It is not suitable for vehicles in underground garages to switch modes with a fast-response system. Therefore, this paper chooses satellite navigation, inertial navigation, and the car system to combine navigation. With the condition that the vehicle can freely travel through indoor and outdoor environments, high-precision outdoor environment navigation is used to provide the initial state of underground navigation. The position of the vehicle underground is calculated by the Dead Reckoning (DR) navigation system. This paper takes advantage of the Extended Kalman Filter (EKF) algorithm to provide two freely switchable navigation modes for vehicles in ground and underground garages. The continuity, robustness, fast response, and low cost of the indoor and outdoor switching navigation methods are verified in real-time systems.

## 1. Introduction

In recent years, as citizen living standard continues to rise, the number of citizens owning cars has increased year by year. According to the data released by the National Development and Reform Commission of China in 2017, the number of private cars has reached 217 million. At present, the ratio of the cars to parking spaces in big cities in China is about 1:0.8. It is about 1:0.5 for small and medium-sized cities, and about 1:1.3 for developed countries. It is conservatively estimated that more than 50 million parking spaces are missing in China. Therefore, the construction of parking space has attracted increased attention. Modern parking lots are gradually developing in the directions of multi-space, multi-function and intelligence [1], since above-ground parking lots have sufficient light and large space and are surrounded by many reference objects so can obtain high-precision position information by Global Navigation Satellite System (GNSS) satellites. On the other hand, underground garages have disadvantages of poor lighting, poor communication quality, crowded parking spaces, and complex building structure [2].

In a complex underground garage, drivers can easily get lost and become unable to find a place to park. At this time, if traffic volume is increased or traffic is blocked, it is easy for accidents such as crashes between vehicles to happen. With narrow space, the garage management system will be brake down. These phenomena will affect the efficiency of the parking lot. Therefore, it is of great practical significance and application value to provide drivers with indoor real-time location to plan reasonable driving routes, according to current traffic [3,4]. Modern satellite navigation technology, the GNSS, can only be used in an outdoor open environment because the satellite signal will be blocked in an indoor environment [5,6]. Some household equipment, such as laser scanners and infrared obstacle avoidance sensors, only support indoor navigation in a small area of closed space. They cannot take into account complex indoor and outdoor switching of scenes smoothly, such as when vehicles enter an underground garage. To solve this problem of compound positioning, it is essential to start with the research of available sensors. Commercial and available sensors, such as laser scanning, laser/fiber gyroscope, visual positioning, wireless positioning, combined Micro-Electro-Mechanical System (MEMS) inertial navigation, etc., are limited for this project. Laser scanning and the fiber gyroscope are relatively expensive and difficult to maintain. They are not suitable for wide-ranging applications, especially for private vehicles. Visual positioning is too complicated to calculate for a car computer and too susceptible to dim light in the garage environment [7], and it is not easy to implement in current car navigation systems. Wireless positioning requires a base station (BS) that arranges a signal in the underground area, because the positioning signal is easily blocked by obstacles such as walls, causing positioning errors in narrow space [8]. Finally, the MEMS integrated navigation technology is not limited by space status and equipment and its price is suitable for popularization in vehicles [9]. In addition, the combination of MEMS inertial technique and satellite navigation can solve the problem of positioning in two different spaces at the same time [10]. This is a better solution for underground garage navigation.

According to different environments, different sensors should be used. When the vehicle is in the open ground environment, the combination of satellite navigation and inertial navigation can greatly improve the accuracy of ground navigation. At the same time, inertial navigation can effectively compensate the positioning error caused by a weak satellite signal in the tunnels, bridges, and the dense area of buildings. When the vehicle goes into an underground environment, the GNSS satellite navigation model cannot work normally [11]. The system is transferred to the inertial navigation plus vehicle navigation mode. Through the DR algorithm [12], the current position of the vehicle can be accurately derived without satellite positioning support. The vehicle navigation system determines whether the navigation runs outdoors based on the PDOP or not [13]. However, the traditional switching method does not respond to the fast and precise requirements of the vehicle. When the vehicle enters a garage entrance, heading deviations will result in an overall offset of the vehicle’s track. Therefore, this paper presents a novel algorithm to switch the status of the navigation system to decrease positioning error. At the stage of a vehicle entering the garage entrance, the vehicle uses the multi-sensor compensation and GNSS heading pre-processing methods to quickly find the state switching point [14]. It determines the accurate initial heading for the subsequent position estimation and improves the accuracy of the vehicle in underground navigation. In the second section, the model establishment, state update, and mode-switching of the basement navigation algorithm are discussed in detail. The third section presents the experiment to verify the theoretical feasibility of the double-mode navigation and analyzes the results. By comparing the environmental characteristics of the two navigation scenarios, the appropriate filter configurations are selected, and the experimental field data is compared to each other for each case. The results illustrate the reliability and robustness of the system from a controllability and practicability point of view.

## 2. System Model Establishment

In this paper, the extended Kalman filter (EKF) is used to update the system state and establish the dynamic system model, the position and velocity of the six-Degree Of Freedom(DOF) rigid body expressed by the inertial coordinates [15]. It is a kinematic model and applicable to any vehicle in the ENU (East-North-Up) Earth fixed frame [16]. Since the IMU sensor is fixed on the vehicle platform, it can only acquire the three-axis Euler angle in the vehicle coordinate system. However, it is not possible to achieve navigation and positioning of the car in the vehicle coordinate system. Therefore, the vehicle coordinate system needs to be rotated in three axes and converted to the geographic coordinate system to realize system positioning. The traditional coordinate system transformation uses the Euler rotation transformation. However, the Gimbal Lock will occur when the Euler rotation transforms in a specific angle. The quaternion transform can effectively avoid the Gimbal Lock phenomenon, because it has one more dimension than the Euler transform. It can also avoid the excessive use of the trigonometric function in the filtering algorithm, making EKF run smoothly in computer programs. In addition, the combination of quaternion and Kalman filtering can use an independent algorithm to find the link among the quaternion, the data of accelerometer, and gyroscope. The converted quaternion participates in the Kalman filter and improves the output of other dimensions, such as position errors. This not only simplifies the number of lines of code in the control system, but also improves the accuracy of vehicle-integrated navigation and positioning. Therefore, the combination of Kalman filter and quaternion in the real-time navigation system is the best choice. The system takes advantage of quaternion rather than the Digital Micro-Circuit(DMC) to represent rotations in 3 dimensions, because quaternion can be used to reduce storage and speed up calculations [17]. The algorithm structure of this paper is shown in Figure 1. The Algorithm 1 consists of two parts: GNSS-integrated navigation system and inertial DR navigation. The two parts are connected by the switching algorithm. According to the quaternion differential equation and quaternion rotation matrix, Formulas (1) and (2) are expressed.
(1)$$\Omega \left(\vec{q}\right)=\left[\begin{array}{cccc} q_{0} & -q_{1} & -q_{2} & -q_{3}\\ q_{1} & q_{0} & -q_{3} & q_{2}\\ q_{2} & q_{3} & q_{0} & -q_{1}\\ q_{3} & -q_{2} & q_{1} & q_{0} \end{array}\right]$$
(2)Reb(q→)=q02+q12−q22−q322q1q2−q0q32q1q3+q0q22q1q2+q0q3q02−q11+q22−q322q2q3−q0q12q1q3−q0q22q2q3+q0q1q02−q12−q22+q32

The system state equation that consists of the current system’s position, velocity, quaternion, and gyro offset 13-dimensional data state variables:(3)X→˙=P→˙V→˙q→˙b→˙w=V→Reb(q→)·a→m+00−gT12Ω(q→)·ω→m−w→m−b→mw→m

The derivative of the system position vector $\dot{\vec{P}}$, the velocity vector derivative $\dot{\vec{V}}$ in the ENU Earth fixed frame, the system attitude quaternion derivative $\dot{\vec{q}}$, the gyroscope static drift derivative ${\dot{\vec{b}}}_{w}$, Reb($\vec{q}$) is the quaternion rotation vector. The system input variable includes the dynamic angular velocity vector ${\vec{\omega }}_{m}$ collected by the gyroscope and the dynamic acceleration vector ${\vec{a}}_{m}$ collected by the accelerometer. The inertial navigation system includes the system white noises ${\dot{\vec{w}}}_{\omega }$ and ${\dot{\vec{w}}}_{a}$ in data acquisition and the three-axis gyroscope bias ${\vec{b}}_{\omega }$.
(4)$$u_{k}=\left\{\begin{array}{c} {\vec{\omega }}_{m}\\ {\vec{a}}_{m} \end{array}\right\}=\left\{\begin{array}{c} \vec{w}+{\vec{w}}_{\omega }+{\vec{b}}_{\omega }\\ \vec{a}+{\vec{w}}_{a}-{\left\{\begin{array}{ccc} 0 & 0 & -g \end{array}\right\}}^{T} \end{array}\right\}$$

The fourth-order Runge–Kutta updates the quaternion state equation [18]:(5)$$K_{1}=f\left(u_{n},X_{n+1}\right)$$
(6)$$K_{2}=f\left(u_{n+\frac{1}{2}},X_{n}+\frac{dT}{2}\cdot K_{1}\right)$$
(7)$$K_{3}=f\left(u_{n+\frac{1}{2}},X_{n}+\frac{dT}{2}\cdot K_{2}\right)$$
(8)$$K_{4}=f\left(u_{n+1},X_{n}+dT\cdot K_{3}\right)$$
(9)$$X_{n+1}=X_{n}+\frac{dT}{6}\left(K_{1}+2K_{2}+2K_{3}+K_{4}\right)$$

The signal collected by the digital system is a nonlinear signal. The state equation is linearized by Taylor expansion and brought into the system measurement matrix $Z_{k}$. In the observation matrix, ${\vec{P}}_{GNSS}$ is the vector of the GNSS three-dimensional position, and ${\vec{V}}_{GNSS}$ is the vector of three-dimensional speed of the GNSS in the local coordinate system [19]. System noise matrix $Q_{k}$ and system measurement noise matrix $R_{k}$ are introduced into the Kalman filter system. All the variables, after state estimation, filter gain, and covariance calculation, finally achieve an updated system state.
(10)$$Z_{k}=\left\{\begin{array}{c} {\vec{P}}_{GNSS}\\ {\vec{V}}_{GNSS} \end{array}\right\}$$
Kalman filter one-step prediction model [16]:(11)$${\bar{X}}_{k}=f\left(X_{k-1},u_{k-1}\right)$$
(12)$$P_{k/k-1}=F_{k}P_{k-1}F_{k}^{T}+\Gamma_{k-1}Q_{k-1}\Gamma_{k-1}^{T}$$ Kalman filter one-step correction model:(13)$$K_{k}=P_{k}H_{k}^{T}{\left(H_{k}P_{k}H_{k}^{T}+R_{k}\right)}^{-1}$$
(14)$$X_{k}={\bar{X}}_{k}+K_{k}\left(Z_{k}-H\left({\bar{X}}_{k}\right)\right)$$
(15)$$P_{k}=\left(I-K_{k}H_{k}\right)P_{k/k-1}$$

The system collects the vehicle driving parameter data from the On-Board Diagnostic (OBD) interface. We collect the instantaneous speed V of the vehicle for the calculation without any influence on the vehicle. There is a certain tilted angle in the process of installation to the vehicle. To correct the installation angles of pitch θ and the roll γ, the installation error caused by the calculation needs compensation. Since the direction of X is determined, the compensation formula is:(16)$$C_{n}^{b}=\left[\begin{array}{ccc} \cos\gamma & \sin\gamma \sin\theta & -\sin\gamma \cos\theta \\ 0 & \cos\theta & \sin\theta \\ \sin\gamma & -\sin\theta \cos\gamma & \cos\theta \cos\gamma \end{array}\right]$$
(17)$$\omega_{zo}=C_{n}^{b}\omega_{zi}$$

The system is installed in the back seat of the vehicle and the direction of the vehicle is determined. The gyroscope only needs to compensate the Z-axis gyroscope for steering control. Therefore, the installation error only needs to consider the pitch and roll.

When the vehicle is driving in the underground garage, although the system is in a semi-enclosed environment, the input $u_{k}$ of the system is not changed. However, the GNSS satellite signal in the observation $Z_{k}$ of the system is completely shielded, so that satellite positioning plus inertial navigation combination cannot work normally. According to the current vehicle experimental conditions, this paper uses the indoor DR navigation algorithm to solve the positioning problem when there is no satellite signal. The system observation is modified to speed vector of the vehicle ${\vec{V}}_{c}$, and the system brings the vector into the Kalman filter by the combination of the vehicle speed and the heading angle:(18)$$Z_{k}=\left\{{\vec{V}}_{c}\right\}$$
(19)$$\theta_{zm}=\int_{0}^{t}\left({\vec{\omega }}_{m}-{\vec{w}}_{\omega }-{\vec{b}}_{\omega }\right)dt$$ The heading angle is obtained by eliminating the system installation error according to Formula (17):(20)$$\theta_{\mathrm{ya}w}=C_{n}^{b}\cdot \theta_{zm}$$ System location update equation:(21)$$L\left(k\right)=L\left(k-1\right)+\int_{k-1}^{k}\frac{V_{N}\left(\tau \right)}{R_{M}+h}d\tau$$
(22)$$D\left(k\right)=D\left(k-1\right)+\int_{k-1}^{k}\frac{V_{E}\left(\tau \right)}{\left(R_{N}+h\right)\cos L}d\tau$$
(23)$$h\left(k\right)=h\left(k-1\right)+\int_{k-1}^{k}V_{D}\left(\tau \right)d\tau$$

The system position is updated according to the speed fluctuation, where $R_{M}$ is the radius of curvature of the meridian circle and $R_{N}$ is the radius of curvature of the circle. $L\left(k\right)$ is the latitude of the system updated, $D\left(k\right)$ is the longitude and the $h\left(k\right)$ is the height above the sea level after the system is updated. The system calculation of the position information of the next moment stems from the position of the previous moment and the three-axis speed integral in the geographic coordinate system. In other words, the vehicle navigation system only needs the current vehicle traveling velocity and its heading. The carrier speed V is decomposed by the heading angle $\theta_{yaw}$, and the position data can be inferred [20].

The heading angle is only calculated by the angular velocity and it can only be obtained from the gyroscope in underground garage [21]. When the vehicle starts up in the garage, the positioning information and heading information of the vehicle cannot be obtained at this time. The state equation $\dot{\vec{X}}$ and the input variable $u_{k}$ remain the same and the geomagnetic vector $\vec{Be}$ is introduced to enter the observation $Z_{k}$ to initialize the heading for the system. The initial position of the system is determined by the last parking position of the vehicle.
(24)$$Z_{k}=\left\{\begin{array}{c} {\vec{V}}_{c}\\ \vec{Be_{c}} \end{array}\right\}$$

The switching navigation mode is the connection part of the two navigation modes, and it is also a relatively difficult part to handle. To judge whether the vehicle is outdoor or not is according to the existence of the GNSS signal, which can be realized as the core thought of the traditional method. However, when the vehicle enters the underground garage, the obstacles such as the glass roof of the garage will weaken the GNSS signal, but it cannot make the signal disappear in short time. As a result, the vague satellite signal causes the position of the vehicle to be greatly deviated. Therefore, the underground garage navigation cannot achieve accuracy based on the existing location data.

**Algorithm 1:** Operational process of the switching algorithm1: Initialize the system vehicle speed, attitude and position2: **for** System loop3:  The system collects relevant experimental data through the data port4:  Update the system attitude quaternion5:  **if** The system is located in the ground environment6:    The system updates the current position and speed by GNSS7:    **if** The system detects the switching flag signal8:      The system shields the GNSS satellite signal9:    **end if**10:  **else** The system is located in underground garage environment11:    The system updates the current position by gyroscope and vehicle speed12:    **if** The system is started in underground garage
      The magnetometer initializes heading of the gyroscope13:    **end if**14:  **end if**15: **end for**

After theoretical derivation and experimental verification, when the current driving speed V is less than half of the average vehicle speed $V_{av}$, the vehicle enters the deceleration state. The vehicle heading data is converted from the GNSS track calculation to the DR algorithm in advance, and the position data continues to be achieved from the GNSS satellite signal [22]. At this moment, the system enters the pre-switching state. If the angle sensor does not detect the angle of entrance of the underground garage or the satellite signal disappears during a period, the system will switch back to the heading angle from the tangent of GNSS track. If the pitch angle of the vehicle $\theta_{c}$ is less than the half of the underground garage inclination angle $\theta_{g}$ minus the equipment installation angle $\theta_{f}$, the state will be signed. Furthermore, if the GNSS satellite signal enters a poor range or the satellite PPS stops counting, the system will also provide a flag for the states. When the five conditions are met at the same time, the current vehicle has entered the entrance of the underground garage. After the angle of entrance is verified by the algorithm, the DR algorithm of the vehicle can run a stable and accurate trajectory in the underground garage.
(25)$$\left\{\begin{array}{c} V\leq \frac{V_{0}+V_{av}}{2}\\ 0<\frac{V_{n}-V_{n+1}}{dT} \end{array}\right.$$
(26)$$\left\{\begin{array}{c} \theta_{c}\leq \frac{\theta_{g}-\theta_{f}}{2}\\ PDOP<7\\ PPS=0 \end{array}\right.$$

## 3. Experimental Verification and Result Analysis

This paper proposes a new method to solve the positioning problem when there is no GNSS satellite signal in the underground garage. The vehicle navigation system platform development board in this work is mainly composed of an inertial measurement unit, communication module, GNSS receiver, car OBD communication module, and ARM core controller. The inertial measurement unit of the navigation system uses MPU9250, which is integrated with triaxial internal gyroscope, triaxial accelerometer, and triaxial magnetometer. UBLOX-M8N is the GNSS receiver and the navigation computer acquires the speed information of vehicle from the car decoding chip. The control core makes use of the ARM-M4 core MCU-STM32. The INS/GNSS combination is a loose combination mode, which is easier to implement in real-time systems. The GNSS positioning uses single-point positioning and the fusion algorithm is Kalman filtering. Figure 2 is the experiment platform. The host computer is mainly responsible for real-time positioning displays and parameter debugging. The software of the host computer is based on map display and it is independently designed for this paper. The image of the software is shown on Figure 3.

All the data stems from the actual vehicle platform in the paper. There are four sets of sampling experiments, which are carried out around the 61 building of Harbin Engineering University. Experiment 1 is PDOP single algorithm-switching control experiment. Experiment 2 is an integrated algorithm-switching experiment. Experiment 3 is a test experiment, which the vehicle starts up in an underground garage, and Experiment 4 is long-time stability verification experiment. All experiments are performed with the same actual experimental platform to ensure stable and reliable experimental data.

The experimental equipment connection diagram is shown in Figure 4. The current system core hardware consists of four parts: nine-axis inertial sensor, car communication model, core controller, and display system. In the system, the GCAN-600 chip acquires the real-time running data of the vehicle through the CAN bus of the automobile OBD interface. In addition, all the obtained data is selected to decode the real-time speed of vehicle. Finally, the result is provided to the navigation computer. The UBLOX series of GNSS receiver acquires navigation data from a satellite antenna located at the top of the vehicle and all the information is sent to the navigation computer after data processing. The core controller STM32 is calculated by the model and the algorithm to obtain the current system real-time posture and position data.

**Experiment 1:** The traditional ground-to-underground switching mode is based on GNSS-PDOP signal strength. When the PDOP signal of the GNSS is out of reasonable range, the IMU will switch from the ground mode to underground mode. However, the direct switching method cannot run smoothly in an underground garage. The GNSS fuzzy signal and the feature of delayed signal leads to huge initial deviation in cutting angle of heading. The underground navigation enters the garage’s DR navigation algorithm based on the wrong heading angle. The overall mistake of the vehicle’s trajectory will be shown (as shown in Figure 5a,c), the blue line is the GNSS and red line for the integrated navigation). Figure 5a is the image of the traditional switching algorithm and the Figure 5c is the actual trajectory from fiber INS/GPS equipment. Compared with the satellite signal positioning accuracy—PDOP value and the vehicle pitch angle—the vehicle enters the garage around 60 s (as shown in Figure 5b). At this time, the PDOP value of the satellite only has a small fluctuation and it cannot be fed back to the navigation computer in time. When the vehicle is completely entered into the garage, the satellite signal responded violently at 70 s. At this moment, the GNSS satellite signal was completely shielded and the PDOP value reached the maximum of 99.9. Compared with the reference fiber INS data, the traditional switching mode will cause the initial heading to be shifted when the vehicle enters the underground garage, which will affect all navigation and positioning data. The hysteresis of such signal measurement cannot be applied to the scene where the vehicle requires rapid response and enters the basement. Therefore, the satellite positioning accuracy PDOP value can only be used as a part of the measurement signal reference, and cannot accurately determine the time point of the garage navigation algorithm switching.

Figure 6 shows the dynamic change of the pitch angle of the vehicle when the vehicle enters the underground garage. This paper compares the Kalman filter algorithm with the output of accelerometer. Because of gravity, the output of the Z-axis accelerometer is similar to the pitch of vehicle. The effect of the EKF algorithm can be obviously seen in this comparison. It can be seen that the pitch angle from the Kalman filter is smoother than the output of accelerometer and the system response speed is different from the output of accelerometer. Because of the mixture of the gyroscope, the EKF diagram reacts more slowly than accelerometer, which indicates the reliability of the modeling algorithm. However, the fluctuation of the pitch angle does not constitute the only evidence of the entrance of the underground garage. Therefore, it is also inaccurate to judge the garage entrance by the pitch angle of vehicle. The vehicle navigation system requires stability and reliability. The vehicle pitch angle data with a large degree of misjudgment can only be used as part of the signal reference measurement. In this case, to accurately obtain the heading angle, multiple sensors are required for complementary data verification.

**Experiment 2:** When the running vehicle uses the combination switching algorithm, the speed, satellite, angle, and acceleration are comprehensively judged in switching point. The track map of the traditional switching algorithm is shown in Figure 5a and the combined switching effect diagram is shown in Figure 7a. Because the speed, the acceleration, and the pitch angle of the vehicle are introduced, the navigation system has achieved the sufficient pre-judgment time before entering the garage. Therefore, the system quickly switches to the DR navigation algorithm after entering the storage buffer. The new method avoids the driving track being brought into the wrong zone by the fuzzy GNSS satellite signal. It improves the navigation accuracy of the DR algorithm directly. Compared with the two experimental groups, Figure 7a and the fiber navigation reference position (Figure 7c), the data of the experimental group 2 is more stable and closer to the high-precision positioning data. The switching flag in Figure 7b is the result of combination switching algorithm. Compared with the PDOP GNSS data in Figure 5b, the switching flag is timely and fast. The heading will not make bias at the point of entering the garage. Figure 8 is the position error waveforms of two experiments. Although the switching time points of the two experiments are not the same, it is obvious that the combination switching algorithm reduces the system position error greatly and the system becomes controlled within a reasonable range. However, due to the heading error of the gyroscope, the system positioning error will also increase, following the accumulation of time.

**Experiment 3:** In addition, this paper also tests the reliable algorithm by long-term vehicle-switching effect, as shown in Figure 9. The vehicle can accurately find the heading in the garage entrance and the corresponding driving trajectory is drawn with MATLAB waveform. The experimental results show that the long-term error-free operation of the system ensures the stability of the algorithm and anti-interference ability. The precision of navigation parameters will not decrease, after working for 20 min. It avoids the phenomenon of the random occurrence of the switching algorithm leading to continuous errors in positioning.

**Experiment 4:** After the experiment of long-term stability, this article continues the related experiment, where the vehicle starts up in the underground garage. The pitch angle of the vehicle remains stable in the garage until the vehicle gets into the garage exit where it generates an elevation angle. Figure 10 shows the variation of real-time pitch angle of the vehicle in the garage. The smaller angle fluctuation shows that the velocity of the vehicle in the garage is low and the acceleration/deceleration is not obvious. The time point of the vehicle going out of the garage is easy to extract. Figure 11 is the track map of vehicle which starts in the garage. After the vehicle is out of the garage completely, the GNSS position signal can be swiftly acquired to match the current position. In addition, the GNSS plus inertial combination mode can be smoothly entered.

## 4. Conclusions

According to the above results, the related experimental data is shown in the Table 1. When taking advantage of the traditional navigation switching mode, the system will judge whether the vehicle is on the ground or not through the satellite positioning accuracy (PDOP). However, there is a certain time delay on the satellite positioning accuracy signal. Its hysteresis will produce a fuzzy signal, so this switching method is not suitable for vehicle positioning which requires a fast response. In view of the problem that the positioning is not correct and switching in time, this paper proposes a combination position switching algorithm. The new algorithm uses a two-step switching mode of pre-predicting and reconfirming, which effectively avoids switching delays and switching errors in normal conditions. In the mathematical statistics of Figure 6, the tilt angle of the underground garage is 8.2°. The system pitch angle variance with the accelerometer Z-axis is 13.89. In contrast, when Kalman filtering is used, the system’s pitch angle is smoother and there is no significant noise. Its variance is only 5.75. It shows that the EKF algorithm perfectly combines the accelerometer fast response with the stability of the gyroscope. Furthermore, the pitch angle from EKF has certain anti-interference ability, which is more suitable for a digital nonlinear system model. From the comparison of Figure 5 and Figure 7, it shows that the traditional judgment method of indoor and outdoor environments based on the satellite PDOP data requires certain reaction time—about 12 s—to switch the system state. This is not suitable for vehicles, and the fast-response speed is necessary for the mobile platform. With the combination switching method, without any other device, the combination of the speed and attitude of the vehicle platform and the GNSS satellite signal can greatly shorten the system switching time to 1 s. In addition, the initial position and heading of vehicle in underground garage are obtained when the system is switched. Therefore, rapid and accurate switching is the guarantee of the accuracy of the positioning of the system in garage. In traditional PDOP switching mode, the position error of the vehicle climbs to 9.1 when the system enters the underground garage. However, when the system position error uses the combination switching method, the position error is only 1.1. It fully illustrates the necessity of the combination of switching algorithm in vehicle navigation field. Figure 9 verifies that the system maintains the accuracy of error-free storage after a long period of driving on the ground, and the controllability that does not interfere with the non-basement environment. Figure 11 proves that when the system is started in an underground garage without satellite signal, the system can still operate accurately, and the outbound storage is connected to the outdoor navigation mode. The system has availability and robustness in multiple environments.

Experiment 1 compares the driving trajectories between the traditional method which is the satellite PDOP switching mode and the fiber inertial navigation. According to the experimental trajectory, the traditional method to judge whether the vehicle enters the underground garage or not has higher delay and error. It is not suitable to use it on the vehicle platform. In the second experiment, the combination switching algorithm is used to draw the driving trajectory and the same fiber inertial navigation trajectory is a contrast to the experiment device. Compared with the data of Experiment 1 and Experiment 2, the combination switching algorithm has higher coincidence with the fiber navigation trajectory and more accurate positioning. To verify the stability of the combination switching algorithm, Experiment 3 tests the effect of the system for 20 min. The result shows that the system runs stably and does not misjudge in non-underground garage areas. Experiment 4 introduces the situation that the vehicle is started in the underground garage and tests the practicability of the system in daily life. The above four experiments indicate that the low-cost navigation device can realize the navigation function and the experimental car is stable and practical in the underground garage.

In this paper, the system hardware equipment is composed of several commercial-grade low-precision inertial navigation devices. The navigation system experimental data accuracy is shown in Table 2. According to the results of fiber inertial and GNSS-INS-integrated navigation system, the position error of integrated navigation is 2 m with the GNSS signal. In the simple inertial navigation algorithm, the position error is 5 m due to the lack of GNSS position correction. It can be seen that the use of the DR inertial navigation algorithm is only suitable for the continuous position-tracking scene indoor environment in a short time. Therefore, the use of such an algorithm is more appropriate for the underground garage.

## Figures and Tables

**Figure 1 sensors-19-01861-f001:**
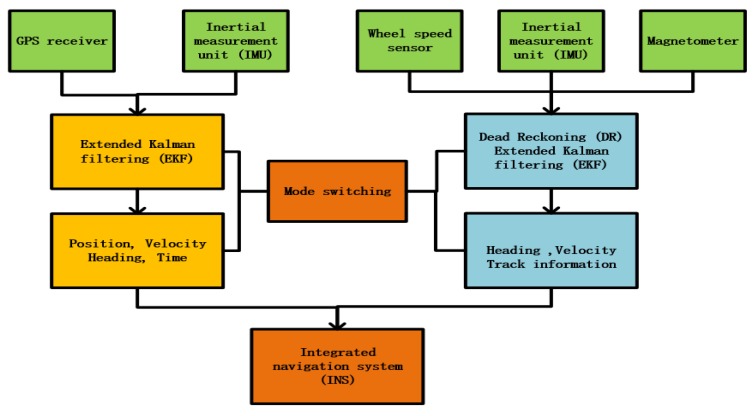
System algorithm structure diagram.

**Figure 2 sensors-19-01861-f002:**
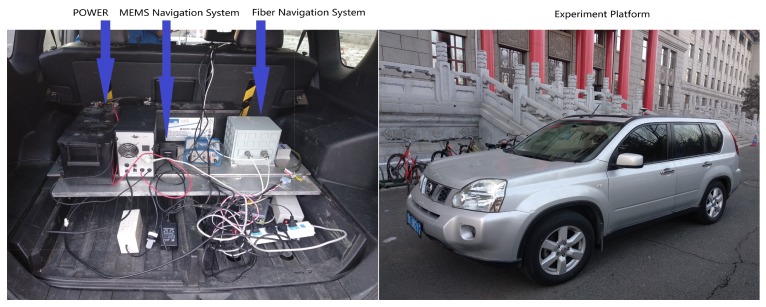
Experiment platform.

**Figure 3 sensors-19-01861-f003:**
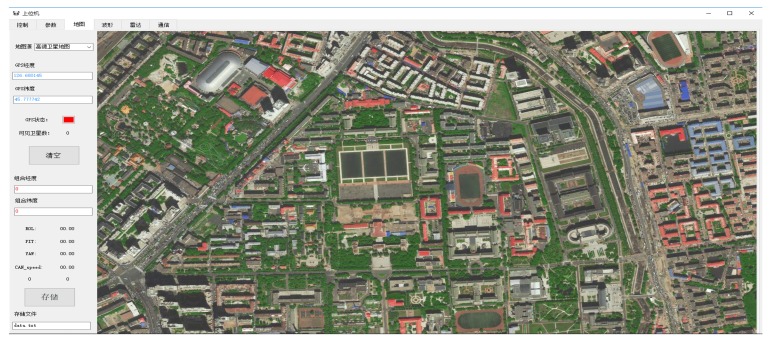
Display system for experiment.

**Figure 4 sensors-19-01861-f004:**
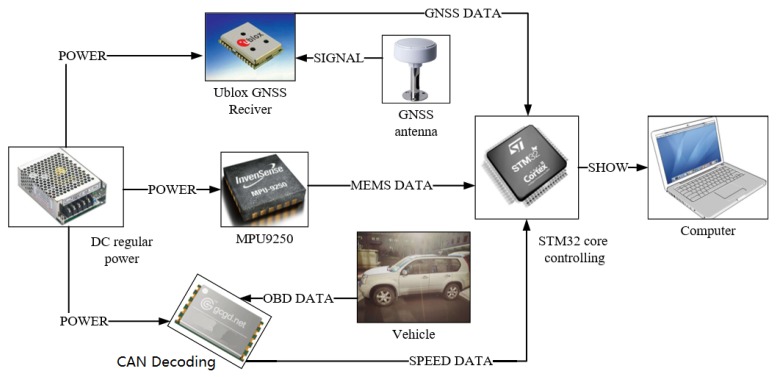
Experimental equipment structure diagram.

**Figure 5 sensors-19-01861-f005:**
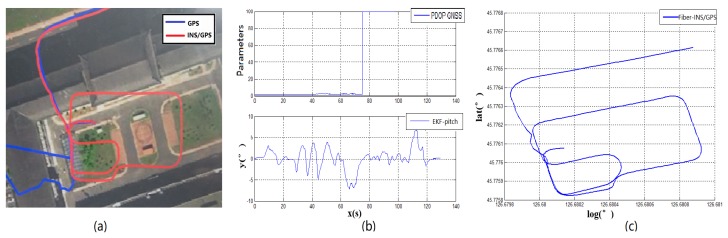
Experiment 1 test result.

**Figure 6 sensors-19-01861-f006:**
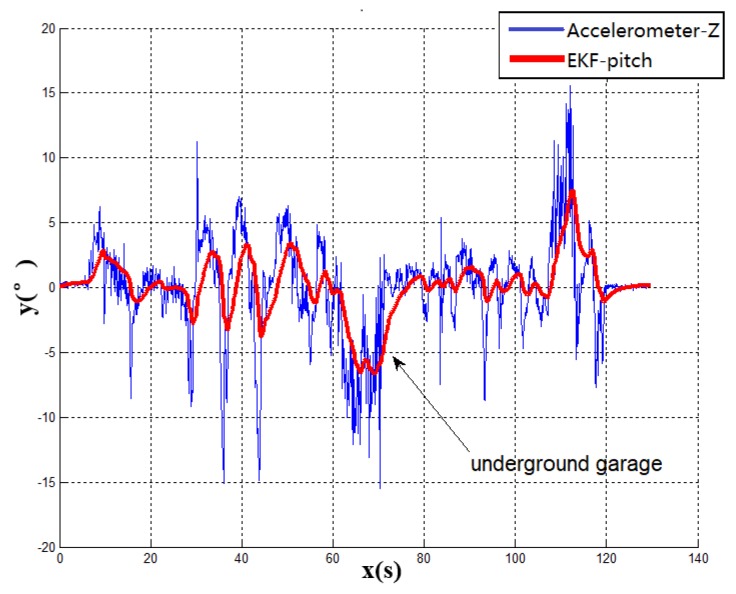
Experiment 1 pitch angle waveform diagram.

**Figure 7 sensors-19-01861-f007:**
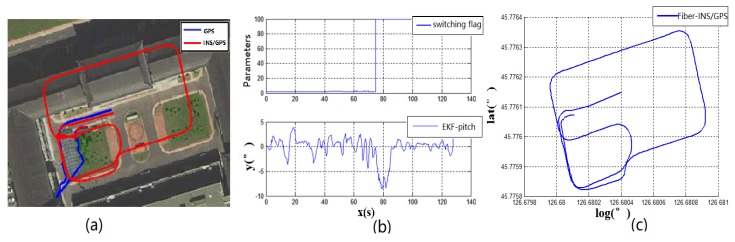
Experiment 2 test result.

**Figure 8 sensors-19-01861-f008:**
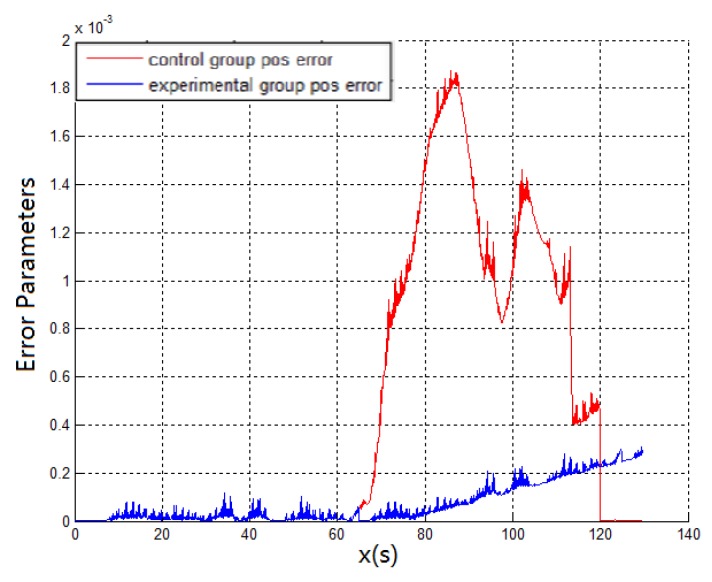
Experiment 1 and Experiment 2 position error.

**Figure 9 sensors-19-01861-f009:**
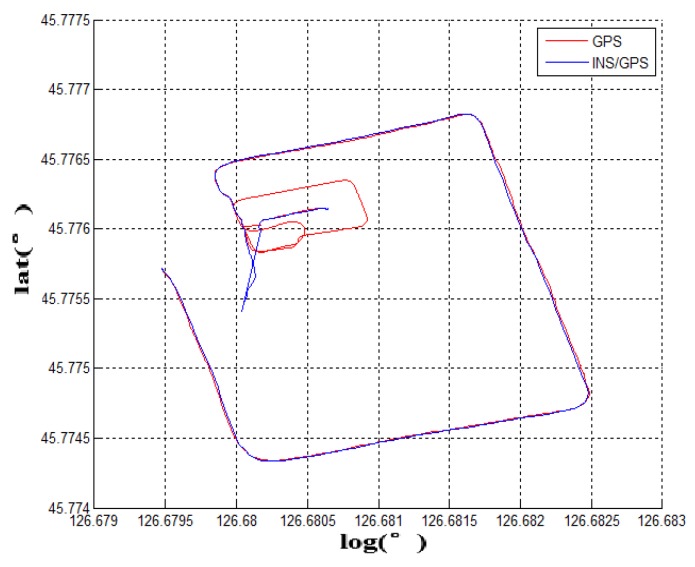
Long-term position trajectory.

**Figure 10 sensors-19-01861-f010:**
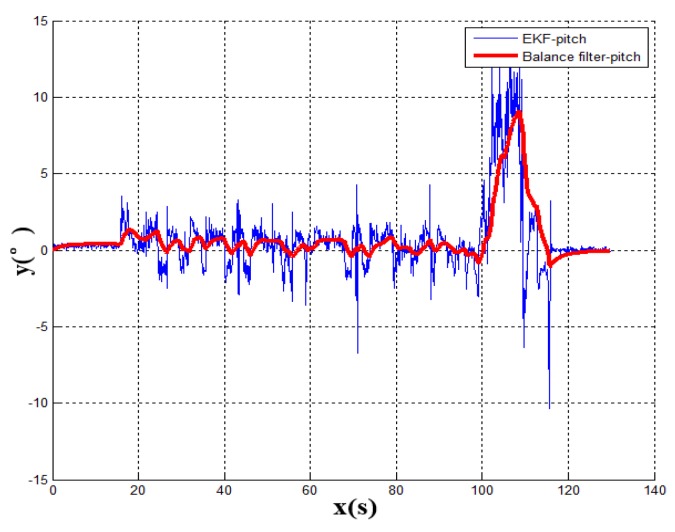
Experiment 4 pitch angle waveform diagram.

**Figure 11 sensors-19-01861-f011:**
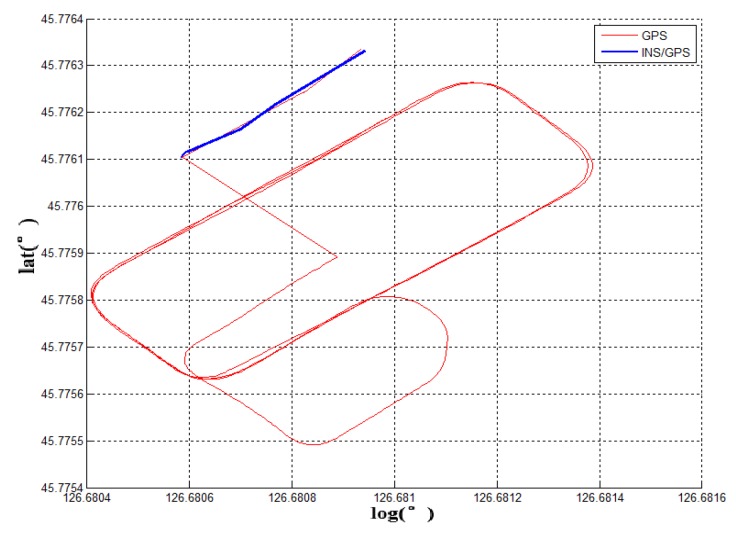
Long-term position trajectory.

**Table 1 sensors-19-01861-t001:** Statistics of system experimental data.

Garage tilt angle	8.2°
Variance of complementary filter	13.89
Variance of Kalman filter	5.75
Response time of traditional switching	12 s
Response time of combination switching	1 s
The average of position error of traditional switching	9.1
The average of position error of combination switching	1.1

**Table 2 sensors-19-01861-t002:** Navigation system experimental data accuracy.

	GNSS-INS	DR-INS(150s)
Attitude	0.5°	1°
Velocity	0.1 m/s	0.5 m/s
Position	2 m	5 m

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
