# Peer review of "A Low-Cost Underground Garage Navigation Switching Algorithm Based on Kalman Filtering"

_sensors, 2019, doi:10.3390/s19081861_

Round 1

Reviewer 1 Report

The topic of this paper is interesting and relevant to the journal scope.

The paper is not very well written. In particular, the first couple of paragraphs are rather confusing and the language seems not to be using the suitable words. 

The authors use the quartenion in section 2 without motivation. This should be introduced, and its importance in the EKF formulation explained. 

The authors present their equipment and then proceed to present a number of experiments, without real motivation or a proper experimental design. It is therefore not clear what each experiments contributes. The authors provide some general figures, from which some visual information can be extracted, but some statistics would be useful to extract some more precise and conclusive information.

The figures are also not well done, as the axis labels and markings, and the legends are not legible. 

The concluding section is very short and shallow. This is the first time some quantitative information is presented.

Minor comments:

- p. 1, l. 1: civilian's: do the authors mean citizens?

- p. 1, l. 10: "as we all know" rather informal, but still style decision

- p. 1, l. 10: "have wide vision"?

- p. 2, 2nd to last line: "repalce"

- Figure 1: there are several typos, e.g. "Extened" (twice) 

- Table 1: the algorithms should be presented as an algorithm, including correct indentation. It should probably also not be called a Table.

Author Response

 REVIEW:

The topic of this paper is interesting and relevant to the journal scope.

The paper is not very well written. In particular, the first couple of paragraphs are rather confusing and the language seems not to be using the suitable words.

The authors use the quartenion in section 2 without motivation. This should be introduced, and its importance in the EKF formulation explained.

The authors present their equipment and then proceed to present a number of experiments, without real motivation or a proper experimental design. It is therefore not clear what each experiments contributes. The authors provide some general figures, from which some visual information can be extracted, but some statistics would be useful to extract some more precise and conclusive information.

The figures are also not well done, as the axis labels and markings, and the legends are not legible.

The concluding section is very short and shallow. This is the first time some quantitative information is presented.

---Response: Thank you for your nice suggestions and comments. They are valuable for improving this paper. According to your insightful advices, we will firstly revise this paper from the beginning to the end to make your suggestions to be fully considered. Specifically, the motivations of algorithm and experimental design would be explained carefully in the updated paper. Then, according to the actual needs, there are four groups of experiments will be designed and explained, and the role significance of each group of experiment will be explained. In addition, the unclear part of the pictures will be re-bold and enlarged. The conclusion part would add the statistical data table, which will make the conclusions to be more persuasive and substantial. Finally, a native English speaker will be invited to improve the overall language of the paper to make the language to be more readable.

2 MAJOR COMMENTS:

- p. 1, l. 1: civilian's: do the authors mean citizens?

---Response: Yes. This is a less used vocabulary. In order to avoid ambiguity the word ‘citizens’ is used here.

- p. 1, l. 10: "as we all know" rather informal, but still style decision

---Response: Thank you for your mention. The informal expression like this would try to be avoided in this paper.

- p. 1, l. 10: "have wide vision"?

---Response: Here, we want to express that the above-ground parking lot have sufficient light and large space, which is suitable for positioning and data acquisition of visual sensors, such as traditional cameras.

- p. 2, 2nd to last line: "repalce"

---Response: This is an incorrect spelling, it should be ‘replace’, thank you for your suggestion.

- Figure 1: there are several typos, e.g. "Extened" (twice)

---Response: This is an incorrect spelling. It should be ‘Extended’, thank you for your suggestion.

- Table 1: the algorithms should be presented as an algorithm, including correct indentation. It should probably also not be called a Table.

---Response: Thank you for your advice on algorithm descriptions. This cannot be called a table. After reading your suggestions, we have read a number of related papers for reference and modified our algorithm description with a standard format. These suggestions make us discover the drawback of the writing format, we will improve the writing skills as fast as possible, thank you.

Thank you again for your suggestions, we hope to learn more from you.

Reviewer 2 Report

   I am very glad to review this paper.

Authors discussed the underground garage navigation switching algorithm. A kind of integrated navigation system, which combines the INS, DR, GPS etc, is researched and developed, and EKF algorithm is adopted to provide two freely-switchable navigation modes for vehicles in the ground and underground garages. And some experiments verified the feasibility of the system and correctness of the algorithm.

1) In the matrix of Equation (14), the element of row 1 and column 2 is wrong, must be sin(g)*sin(q).

2) Please distinguish the specific force from the motion acceleration. The output of accelerometer is specific force, which is equal to the vector difference between the motion acceleration and gravitational acceleration vector. So in the equation (3), “+{0 0 g}”may be “-{0 0 g}”, in the equation (3), “-{0 0 g}”may be “+{0 0 g}”. Please think carefully .

3)  Please explain the meaning (function of observables) of h(Xk), and distinguish altitudes h(k), h(k+1), h , h(tk) from  h(Xk).

4) Roll is denoted by t (above eq.(14)) and q,  please unify. 

5) What is the “direction of X”? In eq.(3) and eq. (5-9), why distinguish x(lowercase) from X(capital), please unify.

6)In page 9, “garage title” ® “garage tilt”, what are the measurement units about values ”8.2”, ”13.89” and “5.75”.

7)In page 10 and 11, “comprehensive” ® “comprehensively”, “¼of the vehicle is” ®¼of the vehicle are”, “more stable and close” ® “more stable and closer”, “reduce and controlled” ® “reduce and be controlled”, “it avods” ®“it avoids”.  Where is Figure 12 and 13? Please recheck.

8)in Abstract , “use efficiency” ® “efficient use”, “join” ® “enter into”?

       This manuscript can be accepted with minor revision. 

Author Response

REVIEW:
I am very glad to review this paper. Authors discussed the underground garage
navigation switching algorithm. A kind of integrated navigation system, which
combines the INS, DR, GPS etc., is researched and developed, and EKF algorithm is
adopted to provide two freely-switchable navigation modes for vehicles in the ground
and underground garages. And some experiments verified the feasibility of the system
and correctness of the algorithm.
---Response: Thank you very much for your insightful comments, which are of great
value for us to improve this manuscript. According to your suggestions, some
shortcomings are revealed in our current research work and we will improve our
research level and achieve more fruitful results in future work. All of your suggestions
have great significance for guiding our research work.
2 MAJOR COMMENTS:
1) In the matrix of Equation (14), the element of row 1 and column 2 is wrong, must
be sin(g)*sin(q).
---Response: Thanks for your valuable attention about this mistake. The updated and
corrected matrix of Equation (14) is shown in the paper. Thanks for your advice.
2) Please distinguish the specific force from the motion acceleration. The output of
accelerometer is specific force, which is equal to the vector difference between the
motion acceleration and gravitational acceleration vector. So in the equation (3), “+{0
0 g}”may be “-{0 0 g}”, in the equation (3), “-{0 0 g}”may be “+{0 0 g}”. Please
think carefully.
---Response: The correct formula of the equation (3) is + {0 0 -g} and the equation (4)
2
should use -{0 0 -g} in ENU Earth fixed frame. The equation (4) is the decomposition
of the input variable of the system. The sensor input variable [am] needs to eliminate
the gravity acceleration -{0 0 -g} to obtain the acceleration vector of the vehicle. It is
convenient for calculating the current posture and heading of the vehicle. In the
formula (3), + {0 0 -g} is used to deduce the derivative of the velocity
three-dimensional vector [V]. First, the three-axis acceleration of the vehicle
coordinate system is converted to the geographic coordinate system, and then the
gravity acceleration [g] needs to be added again. Last, the three-axis speed of the
current vehicle in the geographic coordinate system is obtained after integration.
3) Please explain the meaning (function of observables) of h(Xk), and distinguish
altitudes h(k), h(k+1), h, h(tk) from h(Xk).
---Response: Thank you for taking note of these details in the paper and we really
have vague expression in this point. In Equation 12, h(Xk) should be written as H(Xk),
which is the measurement matrix of the EKF filter. It is the same meaning as the H
matrix in Equations 11 and 13. However, h(k) in Equation 21 represents the solution
expression of the height of the positioning system. In addition, the L(tk), D(tk), and
h(tk) are erroneous expressions and they should be modified, their correct expressions
should be L(k), D(k), and h(k), which would be corrected in the updated paper. The ‘k’
in h(k) and h(k+1) represents a data loop in the system state, ‘k’ and ‘k+1’ represent
the current and next data loop of the system respectively. All the state, except the
GNSS, of the system needs to be updated in one data loop, which is similar with the
clock of the system. In this paper, the system loop ‘k’ is 10ms, which is equal to the
data acquisition clock.
4) Roll is denoted by t (above eq.(14)) and q, please unify.
---Response: Thanks for your suggestion, they are corrected.
5) What is the “direction of X”? In eq.(3) and eq. (5-9), why distinguish x(lowercase)
from X(capital), please unify.
---Response: Thanks for your suggestion, they are corrected.
6)In page 9, “garage title” ® “garage tilt”, what are the measurement units about
values ”8.2”, ”13.89” and “5.75”.
---Response: The unit of 8.2 in page 9 is degree, 13.89 and 5.75 are the statistical
variance of the vehicular system during operation. We think the statistical results are
not as good enough in the article. So, a more clearly result will be presented with a
table in the updated paper.
7)In page 10 and 11, “comprehensive” ® “comprehensively”, “¼of the vehicle is” ®
“¼of the vehicle are”, “more stable and close” ® “more stable and closer”, “reduce
and controlled” ® “reduce and be controlled”, “it avods” ®“it avoids”. Where is
Figure 12 and 13? Please recheck.
---Response: Thank you for your mentions. These mistakes are corrected carefully
3
one by one. The Figure 11 in the conclusion section should be corrected to Figure 9,
and Figure 13 should be corrected to Figure 11.
8)in Abstract, “use efficiency” ® “efficient use”, “join” ® “enter into”?
---Response: Thanks for your suggestions. These mistakes are corrected one by one.

Thank you again for your suggestions, we hope to learn more from you.
